# A multicellular rosette-mediated collective dendrite extension

Li Fan[1], Ismar Kovacevic[1], Maxwell G Heiman[2,3], Zhirong Bao[1]*

[1]Developmental Biology Program, Sloan Kettering Institute, New York, United States; [2]Division of Genetics and Genomics, Boston Children's Hospital, Boston, United States; [3]Department of Genetics, Blavatnik Institute, Harvard Medical School, Boston, United States

**Abstract** Coordination of neurite morphogenesis with surrounding tissues is crucial to the establishment of neural circuits, but the underlying cellular and molecular mechanisms remain poorly understood. We show that neurons in a *C. elegans* sensory organ, called the amphid, undergo a collective dendrite extension to form the sensory nerve. The amphid neurons first assemble into a multicellular rosette. The vertex of the rosette, which becomes the dendrite tips, is attached to the anteriorly migrating epidermis and carried to the sensory depression, extruding the dendrites away from the neuronal cell bodies. Multiple adhesion molecules including DYF-7, SAX-7, HMR-1 and DLG-1 function redundantly in rosette-to-epidermis attachment. PAR-6 is localized to the rosette vertex and dendrite tips, and promotes DYF-7 localization and dendrite extension. Our results suggest a collective mechanism of neurite extension that is distinct from the classical pioneer-follower model and highlight the role of mechanical cues from surrounding tissues in shaping neurites.

DOI: https://doi.org/10.7554/eLife.38065.001

*For correspondence:
baoz@mskcc.org

Competing interests: The authors declare that no competing interests exist.

## Introduction

Morphogenesis of an organism involves long-range coordination among organs. In the nervous system, sensory neurons may grow long dendrites and integrate their sensory endings into target tissues. The guided outgrowth of neurites needs to be coordinated with other neurons as well as surrounding tissues (*Chao et al., 2009*; *Dong et al., 2015*; *Lefebvre et al., 2015*). In particular, many neurites bundle or fasciculate to form nerves. The pioneer-follower model offers a simple developmental mechanism to explain this organization. In this model, the pioneer neuron extends its growth cone first to explore the environment and interact with chemotropic signals and environmental guideposts in order to establish a path to its target. The follower neurons respond to cues from the pioneer (*Tamariz and Varela-Echavarría, 2015*). Multiple mechanisms coordinate the interaction between the pioneer and the followers, such as selective fasciculation (*Hayashi et al., 2014*; *Hutter, 2003*; *Lin et al., 1994*), or juxtaparacrine signals (*Jaworski and Tessier-Lavigne, 2012*).

The amphids are a bilaterally symmetric pair of sensory organs in the nematode *Caenorhabditis elegans*. Each amphid consists of 12 sensory neurons and two glia cells, namely the sheath and the socket cells. The 12 dendrites in an amphid form a sensory nerve, which extends from the neck of the worm where the cell bodies are situated to the nose tip where most of the ciliated endings of the dendrites are exposed to the environment. This structure is highly organized, with the dendrites arranged in a stereotyped order within the bundle (*Yip and Heiman, 2018*). During development, these dendrites grow by retrograde extension, in which the dendritic tips are anchored in place at the embryonic nose while the cell bodies move posteriorly, extending the dendrite behind them (*Heiman and Shaham, 2009*). The anchoring requires DYF-7 and DEX-1, which likely assemble a matrix in the extracellular environment (*Heiman and Shaham, 2009*).

Here, we report how amphid neurons orchestrate collective growth of their dendrites with morphogenesis of the surrounding epidermis. Specifically, we show that the amphid neurons form a multicellular rosette along with the sheath and socket glial cells. The vertex of the rosette, which becomes the dendrite tips, is attached to the anteriorly migrating epidermis and carried to the sensory depression at the developing nose, extruding the dendrites away from the neuronal cell bodies. Abolishing epidermis migration by RNAi of *elt-1*, a key transcription factor required for epidermal fate, abolishes the extension of the amphid dendrites without affecting rosette formation. Molecular localization and loss of function phenotypes suggest that multiple adhesion molecules, including DYF-7, SAX-7/L1CAM, HMR-1/Cadherin and DLG-1/Dlg1 mediate attachment of the rosette vertex to the migrating epidermis. We further show that PAR-6 is localized to the rosette vertex, and promotes DYF-7 localization, as well as the attachment to the epidermis and dendrite extension. Our study reveals a rosette-mediated mechanism for collective neurite outgrowth and nerve formation, in contrast to the classical pioneer-follower model, and highlights a novel role for mechanical cues from the skin in dendrite extension.

## Results

### Amphid neurons form a multicellular rosette before dendrite extension

Using phalloidin staining, we observed a multicellular rosette at the comma stage of *C. elegans* embryogenesis (*Figure 1A*). The age of the embryo and the position of the observed rosette suggested that these could be the amphid neurons. To determine the identity of these cells and understand the developmental function of the rosette, we generated a strain using the *cnd-1*/NeuroD promoter driving PH::GFP to label neurons (*Shah et al., 2017*). 3D, time lapse imaging and cell lineage tracing with a ubiquitously expressed histone::mCherry (*Santella et al., 2014*; *Santella et al., 2010*) showed that the rosette is indeed formed by the amphid neurons (*Figure 1B*). Specifically, the *cnd-1*p::PH::GFP marker labels 9 of the 12 amphid neurons (ADF, ASE, ASG, ASH, ASI, ASJ, AWA, AWB, AWC). The other three amphid neurons are not labeled, but the relative position of their nuclei suggested that they are also part of the rosette. In addition, the sheath and socket cells are labeled and engaged in the rosette. Surprisingly, the marker showed that five other neurons (AIB, AVB, AUA, RIV, URB) are also engaged in the rosette. However, judging by cell shape changes, these non-amphid neurons disengage from the rosette about 40 min later. The significance of this transient engagement is not known.

We found that PAR-6 is localized to the center of the rosette (*Figure 1C*), suggesting that the rosette is a polarized structure. Furthermore, DYF-7, which is required for anchoring the dendrite tips during retrograde extension (*Heiman and Shaham, 2009*), is also localized to the rosette center (*Figure 1D*).

In the next 60 to 90 min, dendrites grow from the rosette center. The dendrite tips migrate anteriorly to the sensory depression at the developing nose. The dendrites stay in a tight bundle, with PAR-6 and DYF-7 localized at their tips (*Figure 1E and F*, *Video 1*). During this period, the cell bodies remain largely stationary (dashed lines in *Figure 1E*). Posterior movement of the cell bodies occurs at later stages (*Figure 1—figure supplement 1*). These results show that the dendrites extend in two distinct steps: first, anterior-directed growth that extrudes the nascent dendrites from the neuronal cell bodies and brings the dendrite endings to the sensory depression at the nose, followed by posterior-directed growth that is concomitant with movement of the neuronal cell bodies.

### The rosette Vertex is carried by the migrating epidermis

At this developmental stage, the epidermal cells are known to migrate anteriorly to enclose the head (*Chisholm and Hardin, 2005*). To examine the relationship between the anterior migration of the epidermal cells and the anterior extension of the amphid dendrites, we conducted 3D, time lapse imaging, using an mCherry::PAR-6 reporter (*Zonies et al., 2010*) to label the rosette vertex and dendrite tips and a DLG-1::GFP to label the junctions in the epidermal cells (*Firestein and Rongo, 2001*; *Totong et al., 2007*). We found that prior to anterior movement of the dendrite tips, the rosette vertex is aligned with the leading edge of the migrating epidermis, specifically the hyp5 cell. As the epidermis migrates anteriorly, an indentation can be observed at the leading edge of hyp5. The dendrite tips are situated in the indentation (*Figure 2A*). Furthermore, the anterior

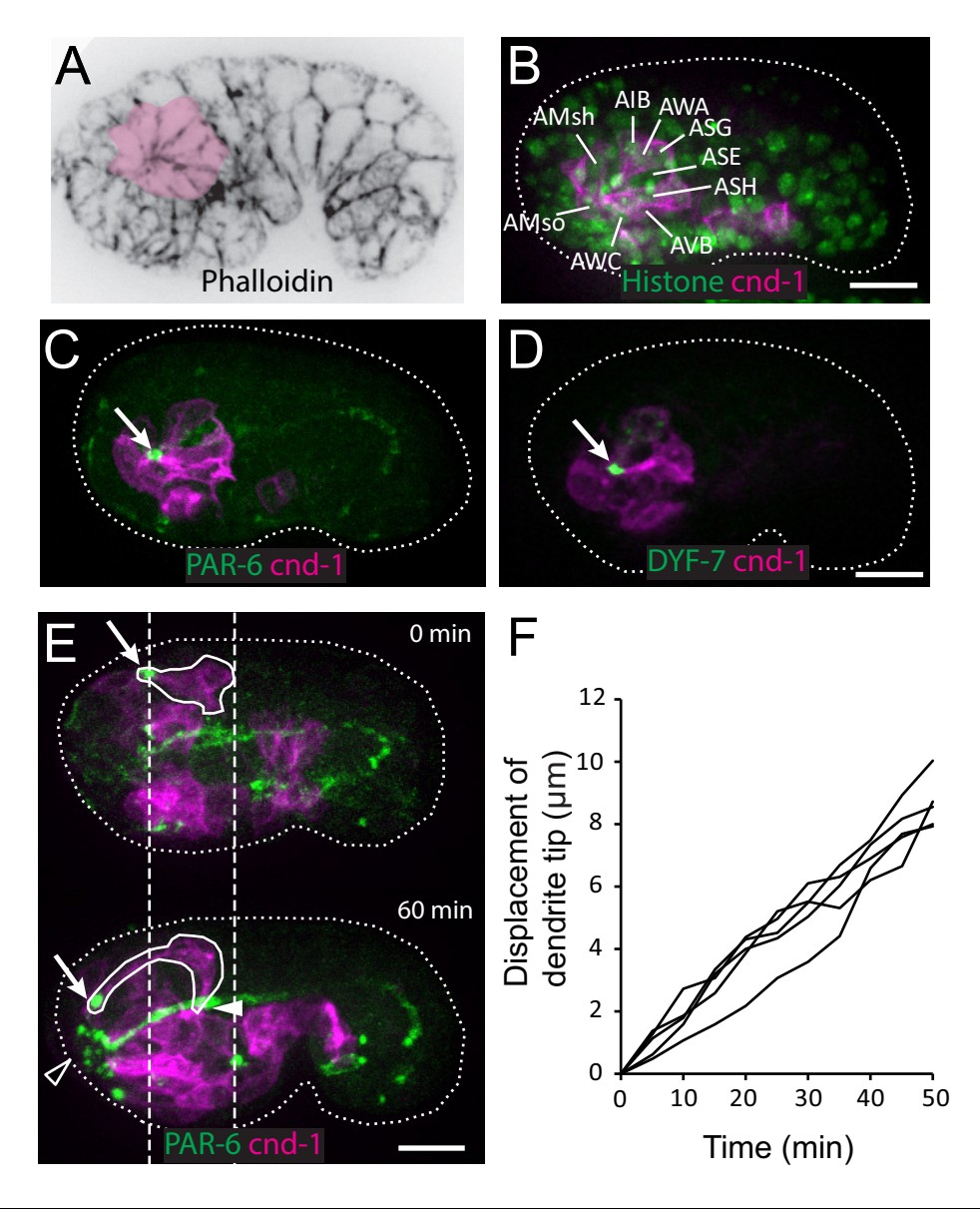

**Figure 1.** Multicellular rosette precedes collective dendrite extension. (**A, B**) Amphid neurons form a multicellular rosette. (**A**) Image of embryo prior to amphid dendrite extension stained with phalloidin. A multicellular rosette is shaded in pink. In this and all subsequent figures, embryos are oriented with anterior to the left. (**B**) *cnd-1*p::PH:: GFP labels amphid neurons in the rosette. Lineage-derived cell identities are shown. (**C, D**) PAR-6 and DYF-7 are localized to the rosette vertex. *cnd-1*p::PH::mCherry labels neurons. (**E**) Amphid dendrite tip migrates anteriorly. PAR-6 accumulates in the dendrite tip. *cnd-1*p::PH::mCherry labels neurons. Vertical dashed lines show initial position of amphid dendrite tip and posterior extent of amphid cell bodies. Arrows indicate dendrite tip. Closed arrowhead indicates amphid axon commissure. Open arrowhead indicates sensory depression. (**F**) Measured displacement of amphid dendrite tips from five embryos during dendrite extension. Scale bars in A-E, 10 μm.
DOI: https://doi.org/10.7554/eLife.38065.002

The following source data and figure supplement are available for figure 1:

**Source data 1.** Displacement of dendrite tip from five embryos.
DOI: https://doi.org/10.7554/eLife.38065.004
**Figure supplement 1.** Retrograde extension in wild-type embryos.
DOI: https://doi.org/10.7554/eLife.38065.003

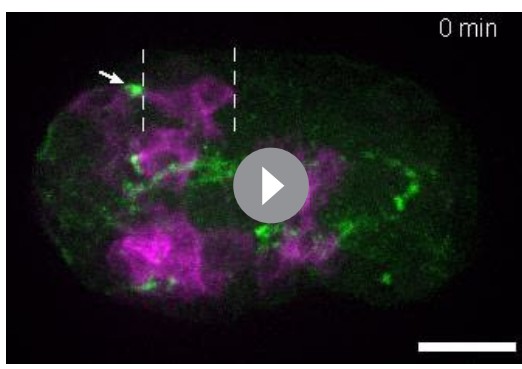

**Video 1.** Amphid dendrite anterior extension. Time-lapse imaging of a wild-type embryo expressing *cnd-1*p::PH::mCherry to label sensory neurons and PAR-6::GFP to label dendrite tips. Dashed lines mark the initial position of the amphid neuron cell bodies at 0 min. The dendrite tip is tracked with an arrow. Scale bar, 10 µm.
DOI: https://doi.org/10.7554/eLife.38065.007

movement of the dendrite tips is correlated with the anterior migration of the leading edge of hyp5 (*Figure 2B*).

These results suggest that the dendrite tips may be physically attached to the leading edge of hyp5 and carried anteriorly. To test this hypothesis, we sought to perturb the anterior migration of the epidermis. ELT-1 is a GATA transcription factor that is necessary to specify epidermal fate (*Page et al., 1997*) and that activates the expression of its target, *lin-26* (*Landmann et al., 2004*). We used *elt-1(RNAi)* to perturb epidermal development, and a *lin-26* promoter driven GFP marker (*Gally et al., 2009*) to label the epidermis and assay the consequence. Expression of the *lin-26*::GFP marker was reduced to different levels depending on the efficacy of RNAi. 24 out of 79 *elt-1(RNAi)* embryos showed high level of expression of *lin-26*::GFP, suggesting the RNAi was not effective in these cases. These embryos exhibited more or less normal anterior migration of the epidermis and the amphid dendrites also developed normally, showing movement that was correlated with the epidermal cells and that brought the dendrite endings to the sensory depression (*Figure 2C*). In contrast, in 29 out of 79 embryos, low levels of *lin-26*::GFP expression were observed, suggesting a moderate effect of RNAi. In these embryos, the epidermis underwent partial anterior migration but stopped before covering the head completely, and the dendrites also showed partial anterior extension, stopping at the position where the epidermis stopped (*Figure 2D*). Intriguingly, the dendrite tips remained coupled to the leading edge of the epidermis even as the epidermis retracted posteriorly after failing to enclose the head in some embryos (dashed lines in *Figure 2D*). Finally, in the embryos with the strongest RNAi effect (26/79), where minimal or no *lin-26*::GFP expression remained, dendrite extension was abolished, even though the rosette formed normally (*Figure 2C*). We were not able to directly observe epidermal cells positions in these embryos due to the strong loss of *lin-26*::GFP, but by inference of RNAi strength and *lin-26*::GFP expression, we assumed that the epidermal migration defect is likely to be more severe than the second group that showed partial migration. These results suggest that the dendrite tips are attached to the leading edge of the epidermis and that dendrite extension requires the anterior migration of the epidermis.

## Multiple adhesion molecules function redundantly to couple epidermal migration to dendrite extension

To identify the molecules that mediate the attachment between the dendrites and the epidermis, we examined the localization of several adhesion molecules during dendrite extension. We generated a construct in which the *dyf-7* promoter drives expression of a modified DYF-7 protein that includes a superfolderGFP (sfGFP) tag on its ectodomain, and that completely rescues the dendrite extension defects of a *dyf-7* mutant (*Low et al., 2019*). As previously described, we found that DYF-7 was expressed by amphid neurons and not in epidermis. As expected from its localization to the rosette vertex (*Figure 1B*), it localized to the dendrite tips (*Figure 3A*). For HMR-1/Cadherin, we used a marker that showed a localization pattern similar to the endogenous pattern as detected by immunostaining and that rescued embryonic lethality of *hmr-1(zu389)* (*Achilleos et al., 2010*). HMR-1 was observed at the dendrite tips in addition to its known localization at epidermal cell junctions (*Figure 3B*). In addition, we examined the localization of SAX-7, a homolog of the vertebrate L1 cell adhesion molecule (L1CAM). SAX-7 has been shown to function redundantly with HMR-1 in blastomere compaction during *C. elegans* gastrulation (*Grana et al., 2010*), and to mediate the interaction between the epidermis and the dendrite of the PVD neuron (*Dong et al., 2013*; *Salzberg et al., 2013*). We used a fosmid-based GFP reporter to examine SAX-7 localization (*Díaz-Balzac et al.,*

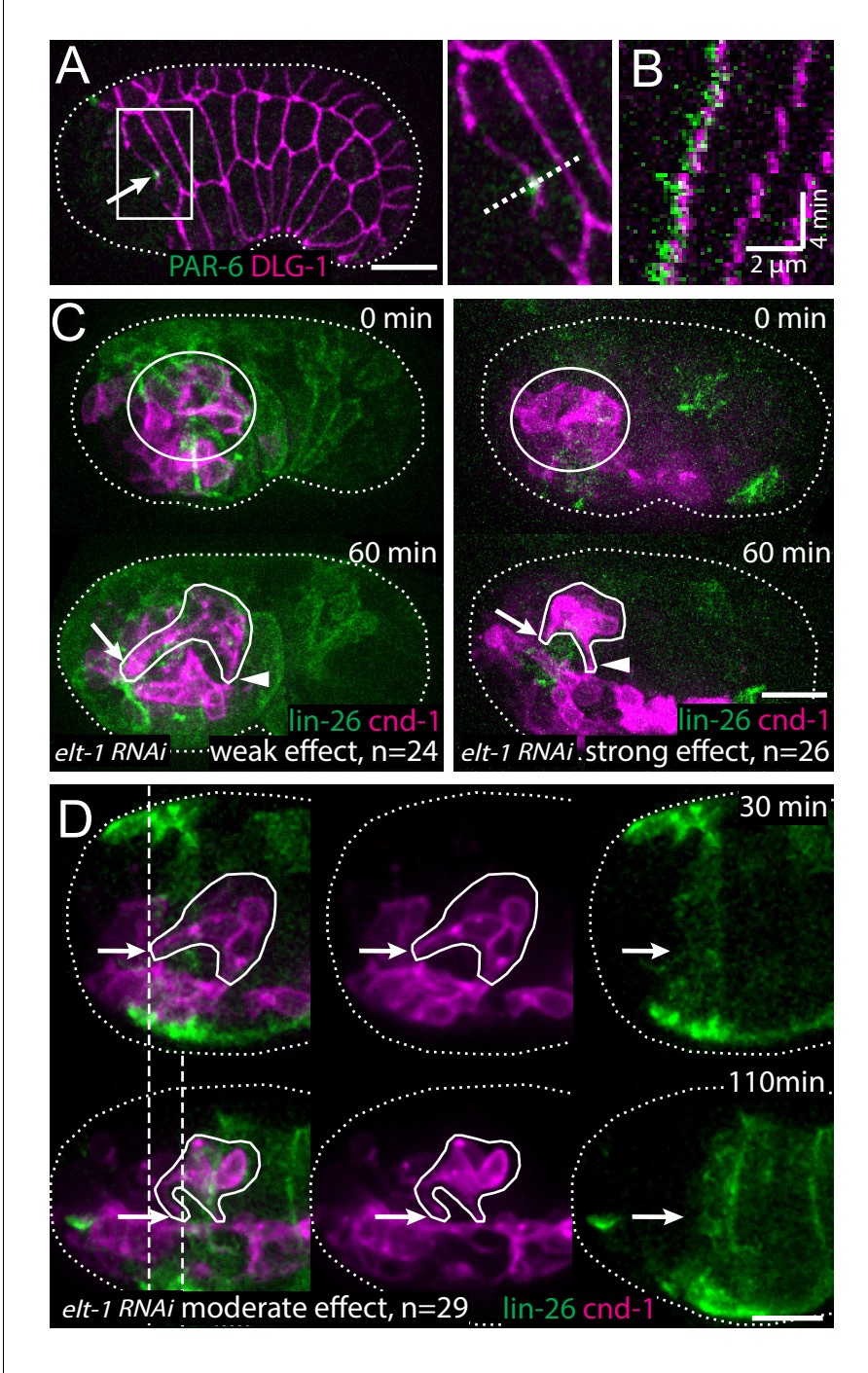

**Figure 2.** The amphid rosette vertex is attached to the epidermis. (**A**) PAR-6-labeled amphid tip (arrows) is localized at the hyp5 leading edge. The apical junction of epidermal cells was labeled by DLG-1::GFP with *xnIs17* transgene. The inset is shown magnified on the right. (**B**) A kymograph generated along the dashed line in the inset in A, showing the correlated anterior movement of the PAR-6 signal and the leading edge of the epidermal cell. (**C**) Example of an *elt-1(RNAi)* embryo with weak RNAi effect which is similar to the wild type (left) and an *elt-1 (RNAi)* embryo with strong RNAi effect and minimal *lin-26*::GFP expression (right). In the latter group dendrites failed to extend (arrow), even though the amphid rosette (circle in the upper panel) and the amphid axon commissure both formed normally (arrowheads). (**D**) A representative *elt-1(RNAi)* embryo with moderate *lin-26*:: GFP expression shows a partial anterior migration of the epidermis (upper panel) followed by posterior retraction. *Figure 2 continued on next page*

*Figure 2 continued*

Dashed lines mark the leading edge of the epidermis before and after the retraction. The amphid dendrite tips follow the leading edge of the epidermis. Arrows mark the position of the dendrite tip. Scale bars in A-D, 10 μm.

DOI: https://doi.org/10.7554/eLife.38065.008

*2016*). We found that SAX-7 is localized to the epidermal cell junctions and along the amphid dendrites (*Figure 3C*). Furthermore, while a DLG-1/Dlg1 reporter used initially in this study (*xnIs17*) showed expression only in the epidermis (*Figure 2A*), we found that a more recently-developed CRISPR tagged DLG-1 (*Heppert et al., 2018*) showed localization on both the apical junctions of epidermal cells and the amphid dendrite tips (*Figure 3D*).

We then asked if these molecules are required for the attachment of the dendrite tips to the epidermis. If a gene is required, its loss of function will cause failure of attachment and result in partial or no anterior extension of the dendrites. In the majority of the *dyf-7(m537)* mutant embryos (84/88), the dendrite tip reached the sensory depression, but detached afterwards when the cell bodies moved posteriorly (*Figure 3—figure supplement 1A*), consistent with the known function of *dyf-7* in the later steps of dendrite extension. However, in a small fraction of *dyf-7(m537)* embryos (4/88), dendrites of one of the amphids extended partially and the tip failed to reach the sensory depression (*Figure 3E*). These embryos reached the 1.5-fold stage normally, and dendrites of the other amphid reached the sensory depression, suggesting that the epidermis migrated normally. This result suggests that DYF-7 also plays a role in the attachment between epidermal cells and dendrite tips during the initial anterior extension of dendrites. DEX-1 has been shown to function together with DYF-7 in dendrite extension (*Heiman and Shaham, 2009*). *dex-1(ns42)* showed similar but weaker phenotypes than *dyf-7(m537)*. In 34/102 of the *dex-1(ns42)* embryos, the amphid dendrite tips detached during the later posterior-directed dendrite extension. 1/102 *dex-1(ns42)* embryos showed defects in the early anterior-directed extension, suggesting that DEX-1 also functions during attachment of dendrite tips to the epidermis (*Figure 3F*). We did not find dendrite extension defects in *sax-7(ky146)*, *hmr-1(RNAi)* or *sax-7(ky146);hmr-1(RNAi)* embryos (*Figure 3F*). Neither *sax-7(ky146)* nor *hmr-1(RNAi)* enhanced the phenotype of *dyf-7(m537)* in the anterior extension of the dendrites. However, in *dyf-7(m537);sax-7(ky146);hmr-1(RNAi)* triple loss of function embryos, we found a significant increase in anterior dendrite extension defects compared to *dyf-7(m537)* (16/88, Fisher's exact test (two-tailed); threshold of significance was adjusted by Bonferroni correction for multiple comparisons, *Figure 3E,F*). In all of the 16 *dyf-7;sax-7;hmr-1(RNAi)* embryos with defective dendrite anterior extension, the embryo developed to the 1.5-fold stage normally and the other amphid tip reached the sensory depression, indicating that epidermal migration in these embryos was not affected. The significantly higher penetrance suggests that DYF-7, SAX-7 and HMR-1 act redundantly in the attachment of dendrite tips to the epidermis. In contrast, 6 out of 72 *dyf-7; sax-7; dlg-1(RNAi)* embryos arrested before reaching the 1.5-fold stage, suggesting that epidermal migration was disrupted. In the remaining 66 embryos that developed to the 1.5-fold stage and beyond, 15 showed a defect in anterior extension of amphid dendrites on one side of the embryo while the dendrite tips of the contralateral side reached the sensory depression (*Figure 3F*). This frequency (15/66, or 23%) is significantly higher than that of *dyf-7(m537)* (4/88, or 5%), suggesting that DLG-1 also functions in the attachment of dendrite tips to the epidermis. We further confirmed the results using *dyf-7 (ns119)* (*Heiman and Shaham, 2009*) and *sax-7(eq1)* alleles (*Wang et al., 2005*) (*Figure 3—figure supplement 1B*).

## PAR-6 functions in amphid dendrite extension

To examine the function of PAR-6 in amphid dendrite extension, we generated *par-6(M/Z)* embryos that are deprived of both maternal and zygotic PAR-6 (*Totong et al., 2007*). In this approach, only a quarter of the embryos imaged in our experiments were expected to be *par-6(M/Z)*. We imaged a total of 89 embryos and found 25 to be *par-6(M/Z)* based on embryonic lethality.

Given the similar localization pattern of PAR-6 and DYF-7, we asked whether PAR-6 functions upstream of DYF-7. In the wild type, DYF-7 was concentrated at the vertex of the amphid rosette (*Figure 4A*) and, following anterior-directed dendrite extension, localized at the dendrite tips (*Figure 4B*). In 20% of the *par-6(M/Z)* embryos (5/25), DYF-7 showed a more diffuse localization

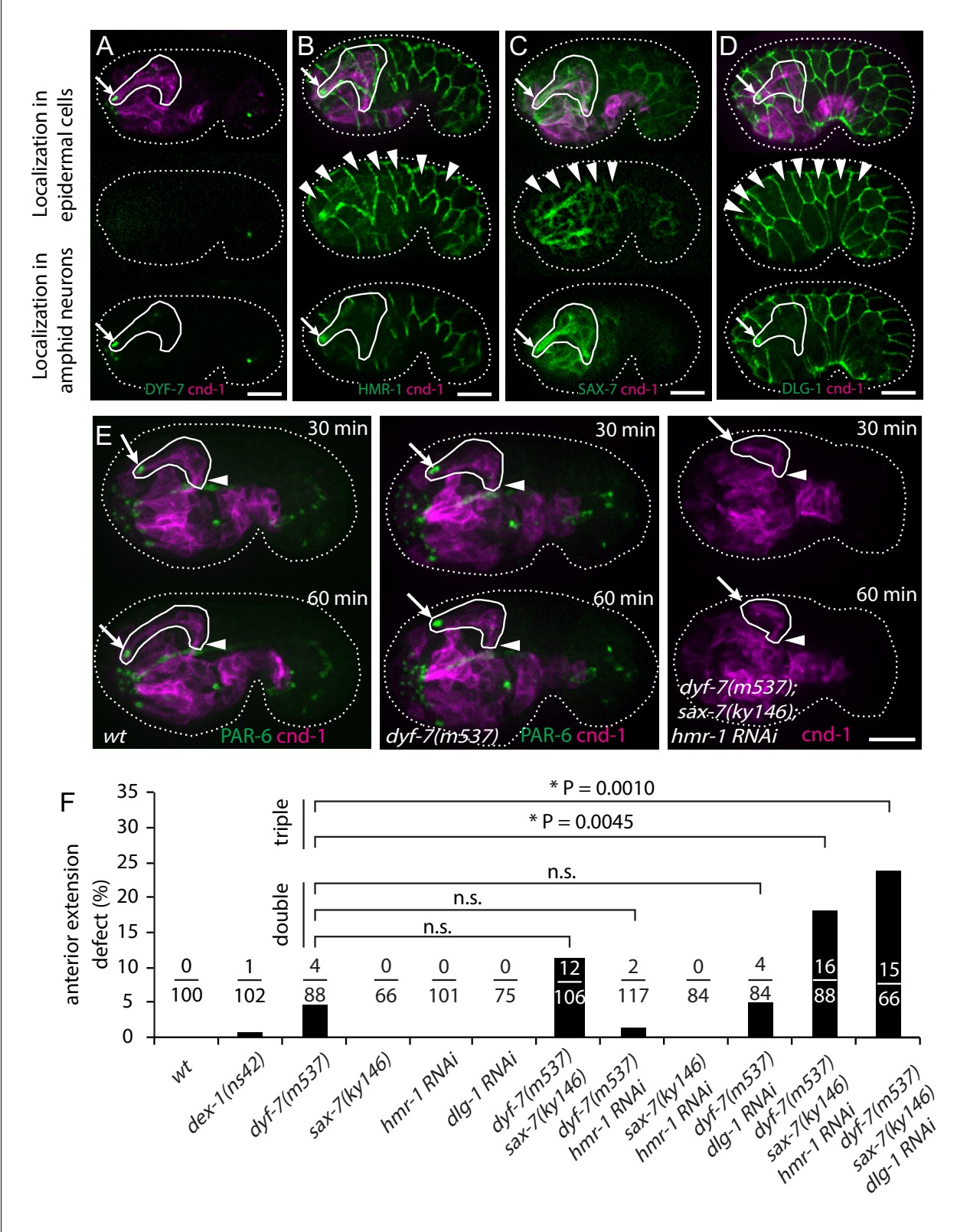

**Figure 3.** Multiple adhesion molecules function redundantly in amphid dendrite anterior extension. (A–D) Localization of DYF-7, HMR-1, SAX-7 and DLG-1 (*dlg-1(cp301)*) in amphid neurons and epidermal cells. Middle panels, superficial focal plane showing localization in epidermal cells. Lower panels, deeper focal plane showing localization in amphid neurons. Arrows indicate amphid tips and arrowheads indicate epidermal cells. (E) Dynamics of dendrite anterior extension in *wt*, *dyf-7(m537)* and *dyf-7;sax-7;hmr-1(RNAi)* triple loss of function. (F) Frequency of anterior extension defects in single,

*Figure 3 continued on next page*

*Figure 3 continued*

double and triple loss of function embryos. Number of amphids scored is indicated. P-values were calculated with Fisher's exact test (two-tailed). Threshold for significance was adjusted by Bonferroni correction to 0.01 for five comparisons. Scale bars in A-E, 10 μm.

DOI: https://doi.org/10.7554/eLife.38065.009

The following source data and figure supplement are available for figure 3:

**Source data 1.** P values in multiple comparisons.

DOI: https://doi.org/10.7554/eLife.38065.005

**Figure supplement 1.** *dyf-7* phenotype and genetic redundancy.

DOI: https://doi.org/10.7554/eLife.38065.010

around the rosette center (*Figure 4A*). The size of the DYF-7::GFP signal at the vertex increased significantly (1.36 ± 0.25 μm in the WT vs 2.27 ± 0.41 μm, student t-test, *Figure 4C*). Furthermore, during later steps of dendrite extension, DYF-7 signal was spread along the dendrites in *par-6(M/Z)* embryos (7/25) (*Figure 4B*). DYF-7 is made as a membrane protein but its extracellular (and functional) domain is cleaved off (*Heiman and Shaham, 2009*; *Low et al., 2019*). Our GFP labels the extracellular domain. It is not yet known if the cleavage occurs before or after DYF-7 localizes to the

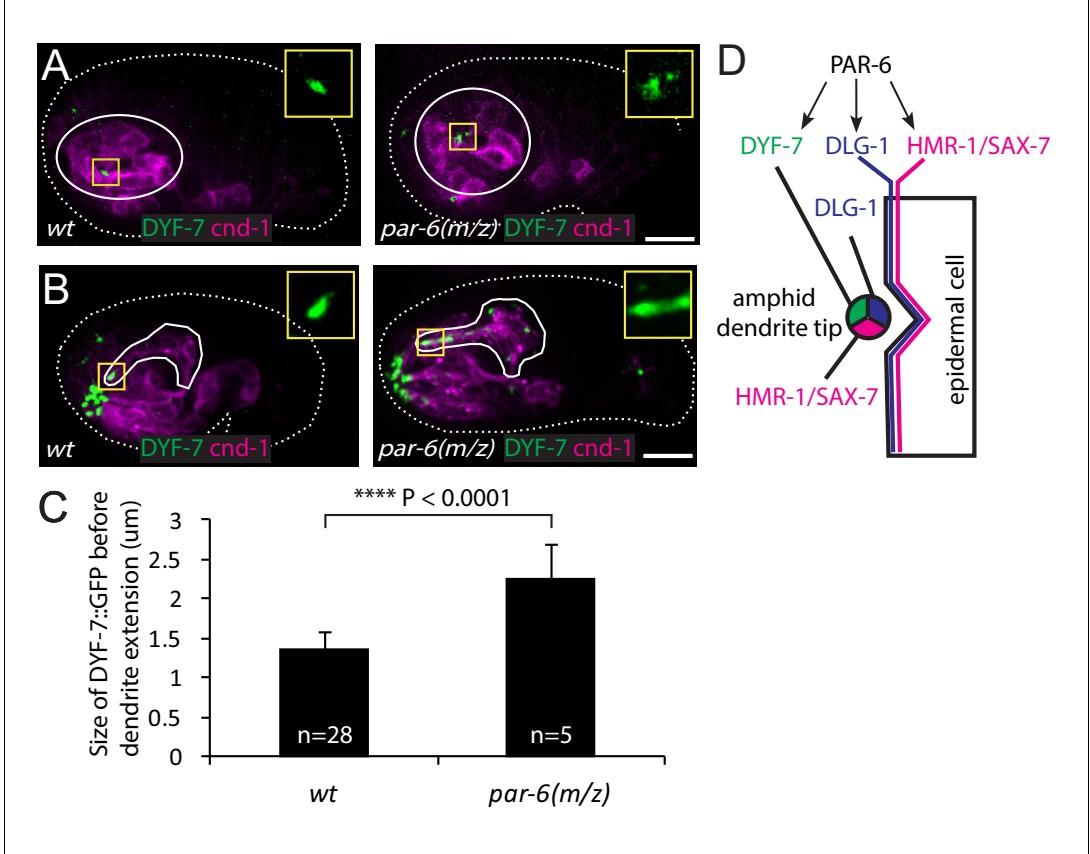

**Figure 4.** PAR-6 regulates localization of DYF-7. (**A, B**) DYF-7 shows more dispersed localization in *par-6(M/Z)*. (**C**) Quantification of the size of DYF-7::GFP in WT and *par-6(M/Z)* embryos before the dendrite tips migrate anteriorly. Data was shown as mean ±SD. P value was calculated with student t-test. (**D**) Schematic model of molecular localization between the amphid dendrite tips and epidermal cell. Scale bars in A-B, 10 μm.

DOI: https://doi.org/10.7554/eLife.38065.011

The following source data and figure supplement are available for figure 4:

**Source data 1.** Size of DYF-7::GFP in *wt and par-6(M/Z)* embryos.

DOI: https://doi.org/10.7554/eLife.38065.006

**Figure supplement 1.** PAR-6 is required for amphid dendrite extension.

DOI: https://doi.org/10.7554/eLife.38065.012

dendrite tips. Nonetheless, these phenotypes suggest that PAR-6 regulates the localization of DYF-7 (*Figure 4D*).

A previous study showed that PAR-6 is required for apical junction formation in epidermal cells by regulating the localization of HMR-1 and DLG-1 (*Totong et al., 2007*). As expected, 17 of the 25 *par-6(M/Z)* embryos arrested early (at 1.5-fold or soon after), indicating early malformation of the epidermis. Among these, five embryos showed partial dendrite extension (*Figure 4—figure supplement 1B*). This phenotype is likely due to the failure of the epidermis to complete its anterior migration. 8 of the 25 *par-6(M/Z)* embryos developed to the 2-fold stage, suggesting that in these embryos epidermal morphogenesis was more or less normal beyond the stage where the dendrite tips normally reach the sensory depression. Among these eight embryos, we found one in which the dendrites showed little extension (*Figure 4—figure supplement 1C*). This result raises the possibility that PAR-6 may play a role in the attachment of dendrite tips to the epidermis. We were not able to image more *par-6(M/Z)* embryos due to technical difficulties.

## Discussion

Our results revealed a mechanism where a multicellular rosette organizes collective neurite outgrowth, which is different from the classical pioneer-follower model in coordinating a group of neurites to grow in the same direction. Multicellular rosettes have been found in diverse organisms and developmental contexts, including *C. elegans* in terms of the assembly of the ventral nerve cord (*Harding et al., 2014*; *Paré et al., 2014*; *Shah et al., 2017*). However, in these previously reported contexts, rosettes are used to orchestrate cell intercalation and collective migration. Dynamic formation and resolution are therefore a defining aspect of those rosettes. In contrast, the amphid rosette remains stable. It is not clear what causes the difference in cellular behaviors between the amphid rosette and the others. Given the localization of PAR-6, HMR-1 and DLG-1, it is possible that the amphid neurons form apical junctions at the rosette center and dendrite tips (*Nechipurenko et al., 2016*). Indeed, electron microscopy of the developing amphid suggests the presence of tight junctions between the dendrites; in the mature structure, tight junctions are present between each of the neurons and the sheath glial cell (*Low et al., 2019*). These junctions define an outward-facing apical surface (*Low et al., 2019*). How DYF-7 interacts with this apical surface would be an interesting question to answer.

We also showed that the initial extension of the amphid dendrites is driven by the migrating skin through physical attachment. Thus, retrograde extension of amphid dendrites occurs in at least two steps: first, neurons attach to the migrating epidermis, which pulls the dendrite endings anteriorly to the developing nose; then, the neuron cell bodies migrate away, pulling the dendrites posteriorly as they move. We refer to this entire process as 'retrograde extension' because dendrite endings remain attached to their target (the epidermis) throughout both steps and dendrite growth is driven by 'stretch' as the distance between the epidermis and the neuronal cell bodies increases – first due to epidermal migration, and then due to neuronal cell body migration. Attachment to a moving target, similar to the first step, has been proposed to underlie axon extension of the M2 pharyngeal neuron in *C. elegans* (*Pilon, 2014*) as well as lateral line 'axon towing' in zebrafish (*Gilmour et al., 2004*). Movement of neuronal cell bodies away from their attachment sites, similar to the second step, has been proposed to underlie olfactory axon development in zebrafish (*Breau et al., 2017*). Thus, the basic mechanisms at play in the amphid may be widely used, but they require careful time-lapse imaging to observe.

Interestingly, the molecular mechanisms appear to be different in terms of the initial attachment of the dendrite tips/rosette vertex to the epidermis during anterior extension and the anchoring of dendrite tips at the sensory depression during the subsequent movement of neuronal cell bodies. In the latter case, *dyf-7* loss of function causes detachment of the dendrite tips from the sensory depression in 100% of the embryos. The same allele, however, only causes a mild phenotype during anterior extension where the dendrite tips detached from the migrating skin in 5% of the embryos. Furthermore, in anterior extension, we showed that SAX-7 and HMR-1 function redundantly with DYF-7 to mediate the attachment of the dendrite tips to the skin. In comparison, recent work suggests that transition zone proteins in cilia function in dendrite extension (*Schouteden et al., 2015*). However, transition zone proteins do not appear to be localized to the amphid dendrite tips until

the 2-fold stage (*Schouteden et al., 2015*), about an hour after the skin-driven anterior extension is complete.

Consistent with the notion that the dendrite tips are attached to different substrates via different molecular mechanisms, we note that during anterior extension, the tips are attached to a specific skin cell called hyp5. However, in the adult, the dendrite tips are embedded between different skin cells, namely hyp2 and hyp3. hyp2/3 are born near the sensory depression and are not part of the anterior skin migration (data not shown). Presumably, the dendrite tips must dissociate from hyp5 and form new relationship with hyp2/3. While this transition remains to be elucidated, it could demarcate the end of the skin-driven anterior extension and the beginning of the later steps of dendrite extension. Thus, the extension of the amphid dendrites involves a surprising level of complexity in term of cellular and molecular mechanisms.

Surrounding tissues play profound roles in neuronal development (*Chao et al., 2009*), and recent work has highlighted intriguing roles for the skin in shaping dendrites. In *Drosophila*, signals from the skin shape Class IV da neurons (*Jiang et al., 2014*; *Meltzer et al., 2016*; *Parrish et al., 2007*), and skin acts to clear debris from degenerating neurons in zebrafish (*Rasmussen et al., 2015*). In *C. elegans*, the epidermis plays a role in controlling synapse density (*Cherra and Jin, 2016*) and positioning (*Shao et al., 2013*). Furthermore, the epidermis plays a critical role in patterning the elaborate dendritic arbor of the PVD neuron. Interestingly, SAX-7/L1CAM functions in this interaction. Specifically, it functions together with MNR-1 in the epidermis to instruct the dendritic pattern of PVD (*Dong et al., 2013*; *Salzberg et al., 2013*). SAX-7 is also required for proper bundling of the amphid neurons (*Sasakura et al., 2005*; *Yip and Heiman, 2018*). In our study, it remains unclear whether SAX-7 acts in the neurons, skin, or both, but the interaction between the amphid dendrites and the epidermis appears to be a simple physical attachment at a focal point defined by the rosette. This example therefore suggests that skin not only provides chemical signals to developing neurons, but can also provide mechanical cues that shape dendrites by physically coupling neurons to epidermal morphogenesis.

An interesting observation regarding the amphid rosette is that key structural features of the mature amphid are already seeded in the rosette before dramatic organ morphogenesis starts. These features include polarization of the neurons and the future dendrite tips at the vertex, engagement of the future dendrite tips with the amphid glia cells, which ultimately ensheath the dendrites, and engagement of the future dendrite tips with the epidermis. Thus, topological features of an organ can be specified early via short range interactions when players are local, before these features are extended over space through development.

Finally, PAR-6 localization to the dendrite tips raises an intriguing question on neuron polarization, given the contrast to its localization and function in vertebrate neurons. In cultured hippocampal neurons, the PAR-6/PAR-3/aPKC complex is enriched at the tip of the future axon. The complex is thought to provide local regulation of the actin cytoskeleton (*Insolera et al., 2011*). A genetic screen in *C. elegans* for mutants that disrupt axon-dendrite polarity in the amphid identified *unc-33*, which encodes a microtubule binding protein called CRMP (*Maniar et al., 2011*). UNC-33/CRMP localizes to the initial segment of the axon and has global effects on microtubule organization in a neuron. The potential function of PAR-6 in the axon-dendrite polarity in the amphid neurons has not been examined. Intriguingly, in *Drosophila* sensory and motor neurons, PAR-6/PAR-3/aPKC localizes to dendrites (*Sánchez-Soriano et al., 2005*). It remains to be seen if this similarity is a coincidence or marks a difference between vertebrate and invertebrate neurons.

## Materials and methods

**Key resources table**

| Reagent type (species) or resource | Designation | Source or reference | Identifiers | Additional information |
|---|---|---|---|---|
| Genetic reagent (*E. coli*) | OP50 | Caenorhabditis Genetics Center | OP50 | |

*Continued on next page*

*Continued*

| Reagent type (species) or resource | Designation | Source or reference | Identifiers | Additional information |
|---|---|---|---|---|
| Genetic reagent (C. elegans) | dex-1(ns42) III | PMID: 19344940 | | |
| Genetic reagent (C. elegans) | dyf-7 (ns119) X | PMID: 19344940 | | |
| Genetic reagent (C. elegans) | dyf-7 (m537) X | PMID: 19344940 | | |
| Genetic reagent (C. elegans) | sax-7(ky146) IV | Caenorhabditis Genetics Center | CX2993 | |
| Genetic reagent (C. elegans) | sax-7(eq1) IV | Caenorhabditis Genetics Center | LH81 | |
| Genetic reagent (C. elegans) | par-6(zu170) I | Caenorhabditis Genetics Center | FT36 | |
| Genetic reagent (C. elegans) | par-6(tm1425)/hIn1 [unc-54(h1040)] I | Caenorhabditis Genetics Center | JJ1743 | |
| Genetic reagent (C. elegans) | unc-101(m1) I | Caenorhabditis Genetics Center | DR1 | |
| Genetic reagent (C. elegans) | zbIs3[cnd-1p::PH::GFP] | PMID: 28441532 | | |
| Genetic reagent (C. elegans) | ujIs113[pie-1p::mCherry::H2B; nhr-2p::mCherry::HIS-24 -let-858UTR; unc-119(+)] II | Caenorhabditis Genetics Center | JIM113 | |
| Genetic reagent (C. elegans) | itIs1024[par-6::PAR-6::GFP] | Dr. Kenneth Kemphues | KK1024 | |
| Genetic reagent (C. elegans) | zyIs36[cnd-1p::PH::mCherry; myo-2p::mCherry]X | PMID: 28441532 | | |
| Genetic reagent (C. elegans) | mcIs40 [lin-26p::ABDvab-10::mCherry; myo-2p::GFP] | Caenorhabditis Genetics Center | ML916 | |
| Genetic reagent (C. elegans) | xnIs17[dlg-1::GFP; rol-6(su1006)] | Caenorhabditis Genetics Center | FT63 | |
| Genetic reagent (C. elegans) | axIs1928[mCherry::PAR-6] | Caenorhabditis Genetics Center | JH2648 | |
| Genetic reagent (C. elegans) | dlg-1(cp301[dlg-1:: mNG-C13xFlag])X | Caenorhabditis Genetics Center | LP598 | |
| Genetic reagent (C. elegans) | ntIs1[gcy-5::GFP] | Caenorhabditis Genetics Center | OH3192 | |
| Genetic reagent (C. elegans) | oyIs44[odr-1::RFP + lin-15(+)]V | Caenorhabditis Genetics Center | PY2417 | |
| Genetic reagent (C. elegans) | zuIs43[pie-1::GFP::PAR-6::ZF1; unc-119(+)] | Caenorhabditis Genetics Center | JJ1743 | |
| Genetic reagent (C. elegans) | ddIs290[sax-7::TY1::EGFP:: 3xFLAG(92C12); unc-119(+)] | Caenorhabditis Genetics Center | TH502 | |
| Genetic reagent (C. elegans) | xnIs96 [hmr-1p::HMR-1::GFP:: unc-54 3'UTR; unc-119(+)] | Caenorhabditis Genetics Center | FT250 | |
| Genetic reagent (C. elegans) | kyIs4[ceh-23-unc-76-gfp::lin-15]X | Caenorhabditis Genetics Center | CX2565 | |
| Genetic reagent (C. elegans) | hmnEx149[dyf-7p::DYF-7 (ZP-sfGFP)-mCherry; rol-6(su1006)] | DOI: 10.1101/393850 | | |
| Genetic reagent (E. coli) | elt-1(RNAi) expressed in HT115 (DE3) | Source BioScience | C. elegans RNAi collection (Vadel) | RNAi Bacteria, Clone ID: 10019-B-8 |
| Genetic reagent (E. coli) | hmr-1(RNAi) expressed in HT115 (DE3) | Source BioScience | C. elegans RNAi collection (Ahringer) | RNAi Bacteria, Clone ID: I-5F23 |

*Continued on next page*

*Continued*

| Reagent type (species) or resource | Designation | Source or reference | Identifiers | Additional information |
|---|---|---|---|---|
| Genetic reagent (*E. coli*) | *dlg-1(RNAi)* expressed in HT115 (DE3) | Source BioScience | *C. elegans* RNAi collection (Ahringer) | RNAi Bacteria, Clone ID: X-8A08 |
| Software, algorithm | Fiji | https://fiji.sc/ | | |

## Worm strains and genetics

Worms were maintained at room temperature on nematode growth media (NGM) plates seeded with OP50 bacteria as previously described (*Brenner, 1974*), except for RNAi experiments. N2 Bristol was used as the wild-type strain. Strains used in this study are listed in the key resource table.

## RNAi

RNAi experiment was performed using the standard feeding method (*Kamath et al., 2003*; *Rual et al., 2004*). For *elt-1(RNAi)* and *hmr-1(RNAi)*, L1 hermaphrodites were placed on RNAi plates and embryos were harvested by cutting the adults 2 days later. For *dlg-1(RNAi)*, L4 hermaphrodites were grown for 1 day before embryos are collected. The clones of 10019-B-8 from Vidal RNAi library, I-5F23 and X-8A08 from Ahringer RNAi library were used to target *elt-1*, *hmr-1* and *dlg-1*, respectively.

## Microscopy

Embryos were collected and mounted as previously described (*Bao and Murray, 2011*). Briefly, embryos were collected by cutting gravid hermaphrodites in a droplet (20 μL) of M9 buffer (3 g $KH_2PO_4$, 6 g $Na_2HPO_4$, 5 g NaCl, 1 ml 1 M $MgSO_4$, per liter $H_2O$). Embryos at the 2 or 4 cell stages were transferred to a droplet (1.5 μL) of M9 containing 20 μm polystyrene beads on 24 × 50 mm coverglass. An 18 × 18 mm coverglass was laid on top and sealed using melted Vaseline. Images were acquired on a spinning disk confocal microscope (Quorum Technologies, Puslinch, Canada) comprising a Zeiss Axio Observer Z1 frame. An Olympus UPLSAPO 60x objective was used with a thread adapter (Thorlabs, Newton, NJ) to mount on the Zeiss body. The timing of major developmental events was used to check for phototoxicity during 3D time-lapse imaging.

## Acknowledgments

The authors would like to thank Shai Shaham for reagents and discussions, Daniel Colon-Ramos and Songhai Shi for advice on the manuscript, and members of the Bao lab for discussions and help. This work was supported by NIH grants (R01 GM097576 and R24 OD016474 to ZB, R01 GM108754 to MGH) and the MSK Cancer Center Support/Core (P30 CA008748). Some strains were provided by Dr. Jeremy Nance, Dr. Kenneth Kemphues, Dr. Shohei Mitani (National Bioresource Project), and the CGC, which is funded by the NIH Office of Research Infrastructure Programs (P40 OD010440).

## Additional information

### Funding

| Funder | Grant reference number | Author |
|---|---|---|
| National Institutes of Health | R01 GM097576 | Zhirong Bao |
| National Institutes of Health | R24 OD016474 | Zhirong Bao |
| National Institutes of Health | R01 GM108754 | Maxwell G Heiman |

The authors declare that the funders had no role in study design, data collection and interpretation, or the decision to submit the work for publication.

## Author contributions
Li Fan, Conceptualization, Data curation, Formal analysis, Investigation, Visualization, Writing-original draft, Writing-review and editing; Ismar Kovacevic, Conceptualization, Data curation, Formal analysis; Maxwell G Heiman, Conceptualization, Resources, Funding acquisition, Writing—review and editing; Zhirong Bao, Conceptualization, Supervision, Funding acquisition, Writing—original draft, Writing—review and editing, Formal analysis

## Author ORCIDs
Li Fan [ID] https://orcid.org/0000-0003-1780-6919
Maxwell G Heiman [ID] https://orcid.org/0000-0002-2557-6490
Zhirong Bao [ID] http://orcid.org/0000-0002-2201-2745

## Decision letter and Author response
Decision letter https://doi.org/10.7554/eLife.38065.015
Author response https://doi.org/10.7554/eLife.38065.016

## Additional files

### Supplementary files
• Transparent reporting form
DOI: https://doi.org/10.7554/eLife.38065.013

### Data availability
All data generated or analysed during this study are included in the manuscript and supporting files.

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
