## [Decision Letter]

Thank you for submitting your article "A multicellular rosette-mediated collective dendrite extension" for consideration by *eLife*. Your article has been reviewed by two peer reviewers, and the evaluation has been overseen by Oliver Hobert as the Reviewing Editor and Eve Marder as the Senior Editor. The reviewers have opted to remain anonymous.

There was a general consensus among everyone that this paper reports on exciting and interesting results. However, the paper does require some specific revisions that clarify several points, as raised by the reviewers. All these points are important but should not present a problem in addressing.

*Reviewer #1:*

In this manuscript Fan et al. describe early aspects of amphid dendrite development in *C. elegans*. A bundle of 12 sensory dendrites belonging to the set of amphid neurons in the head of the animals was previously shown by Heiman and Shaham in 2009 to develop by a mechanism termed retrograde dendrite extension. In their model, the tips of the dendrites attached to the nose of the animals in a manner dependent on two proteins with similarity to extracellular sperm and egg proteins (DEX-1 and DYF-7). These proteins were suggested to form a specialized extracellular matrix, which served as attachment for the dendrite to the epidermis. Subsequent growth of the animal and migration of the sensory neuron cell bodies in a posterior direction then results in retrograde extension of the attached dendrites. In the current paper Fan et al. extend this model and refine it in several ways. The authors show that the amphid neurons first form a rosette the vertex of which forms the developing dendrites. Second, several proteins, including the DYF-7 zona pellucida protein (previously shown to be important for retrograde extension), the HMR-1/Cadherin and the SAX-7/L1CAM cell adhesion molecules and PAR-6 localize to the vertex/tips of this rosette. Third, the movement of the vertex correlates with epidermal migration and is dependent on correctly specified epidermis, suggesting that the anteriorly migrating epidermal cell pulls the dendrites initially towards the tip of the nose. This anterior dendrite extension, already noted by Heiman and Shaham, precedes anterior migration of the neuronal cell bodies and retrograde extension of the amphid sensory dendrites. Further experiments suggest that localization of DYF-7 is dependent on the polarity protein PAR-6 and that the cell adhesion molecules SAX-7 and HMR-1 may act redundantly with DYF-7 to mediate the attachment of the vertex to the migrating epidermal cell.

Overall this is a potentially significant paper for a number of reasons. First, it provides another example of how the epidermis shapes dendrite patterning. Second, the function of the epidermis appears primarily mechanical through attachment of the vertex that is pulled behind the migrating cell rather than through some signaling mechanism. Third, a polarity protein (PAR-6) may be involved in localizing cell adhesion molecules to the leading tip of dendrites in the vertex. Fourth, several cell adhesion molecules function redundantly to potentially mediate this interaction between the epidermis and the dendrite tips. A further strength of the manuscript is the high resolution time lapse imaging of early embryonic processes of neuronal development in *C. elegans*, an aspect that remains, with few exceptions, significantly understudied. A weakness of the paper is that the focus of action for the cell adhesion molecules (HMR-1 and SAX-7) has not been established, and is only inferred from protein localization (which may not allow identification of the cellular source). The site of action for these factors could be important, because the cell adhesion molecules could constitute part of the receptor (complex) on either the dendrites or the epidermis, which interacts with the DYF-7 extracellular matrix protein. Regardless of possible physical interactions between these proteins, a few simple rescue experiments could allow the authors to substantially refine their model figure where, at this stage, the localization of HMR-1 and SAX-7 remains somewhat diffuse as it pertains to the phenomenon of dendrite extension by epidermal pull.

Other points:

• Subsection “PAR-6 functions in amphid dendrite extension”. The conclusion that PAR-6 is required for attachment of dendrites to the epidermis rests on a single embryo. Moreover, the genotype of this one embryo is not clear and only inferred based on the assumption that all dead embryos are par-6 mutant embryos. I would consider this rather weak evidence, although maybe promising.

• The absence of an analysis of *dex-1* is rather curious, given the known role of this gene with *dyf-7* in retrograde dendrite extensions.

• The authors make it sound as if the mechanism they describe goes against the previous model, but it really only refines the initial aspects of this process (third paragraph of subsection “Amphid neurons form a multicellular rosette before dendrite extension”).

*Reviewer #2:*

Bao and colleagues studied the initiation of sensory dendrites in vivo using the *C. elegans* amphid as a system. They described a surprising and exciting finding of a rosette structure where the dendrites of many neurons initiate. The rosette correlates and most likely is organized by a subcellular adhesion structure between the epidermal cell and the developing neurites. The migration of the epidermal cells drag the dendrite along which helps the first step of dendrite extension. The observation of the rosette is striking, original and extremely interesting. This observation provides an example of in vivo dendrite outgrowth that has not been described before. However, the paper is preliminary due to the incompletely characterization of this phenomenon. Several parts of the story can be improved.

1) Figure 2 needs to be quantified. In the description of this figure, it was not immediately clear whether the anterior migration of the epidermal cell is normal or not. This needs to be carefully investigated and quantified.

2) Figure 3, in the triple mutants, is the anterior migration of the epidermal cells affected or is it just the dendrite extension that is defective?

3) For Figure 4, is there a statistical test that the authors can do to test the significance of the various phenotypic categories? It seems that the *par-6* results are very hard to interpret despite the very nice localization. The loss of function analyses is hampered by early defects and very low penetrance of the dendrite extension defect. I suggest removing this part of the data.

4) For the second part of Figure 4, while the area of DYF-7 is detectably larger in the *par-6* mutants compared with the control, the DYF-7 is nevertheless localized to the tip of the dendrite in the mutant. It is not clear what this phenotype really means.

[Editors' note: further revisions were requested prior to acceptance, as described below.]

Thank you for resubmitting your work entitled "A multicellular rosette-mediated collective dendrite extension" for further consideration at *eLife*. Your revised article has been favorably evaluated by Eve Marder (Senior Editor), a Reviewing Editor, and two reviewers.

The manuscript has been improved but there are some remaining issues that need to be addressed before acceptance, as outlined below:

Major:

– Please subject your data to more complete statistical analyses. For example, in Figure 3, the comparisons of the single, double and triple mutants to wild type animals are missing. The analyses should also include corrections for multiple comparisons.

– While the reviewers agreed that rescue may be difficult, they (and the editors) feel strongly that it is not sufficient to derive conclusions from single allele usage, particularly if no rescue is provided. Hence, you are asked to generate a *dyf-7;sax-7* double mutant using different alleles from different labs and test this strain with and without RNAi against e.g. *hmr-1* or *dlg-1*.

---

## [Author Response]

Reviewer #1:In this manuscript Fan et al. describe early aspects of amphid dendrite development in C. elegans. A bundle of 12 sensory dendrites belonging to the set of amphid neurons in the head of the animals was previously shown by Heiman and Shaham in 2009 to develop by a mechanism termed retrograde dendrite extension. In their model, the tips of the dendrites attached to the nose of the animals in a manner dependent on two proteins with similarity to extracellular sperm and egg proteins (DEX-1 and DYF-7). These proteins were suggested to form a specialized extracellular matrix, which served as attachment for the dendrite to the epidermis. Subsequent growth of the animal and migration of the sensory neuron cell bodies in a posterior direction then results in retrograde extension of the attached dendrites. In the current paper Fan et al. extend this model and refine it in several ways. The authors show that the amphid neurons first form a rosette the vertex of which forms the developing dendrites. Second, several proteins, including the DYF-7 zona pellucida protein (previously shown to be important for retrograde extension), the HMR-1/Cadherin and the SAX-7/L1CAM cell adhesion molecules and PAR-6 localize to the vertex/tips of this rosette. Third, the movement of the vertex correlates with epidermal migration and is dependent on correctly specified epidermis, suggesting that the anteriorly migrating epidermal cell pulls the dendrites initially towards the tip of the nose. This anterior dendrite extension, already noted by Heiman and Shaham, precedes anterior migration of the neuronal cell bodies and retrograde extension of the amphid sensory dendrites. Further experiments suggest that localization of DYF-7 is dependent on the polarity protein PAR-6 and that the cell adhesion molecules SAX-7 and HMR-1 may act redundantly with DYF-7 to mediate the attachment of the vertex to the migrating epidermal cell.Overall this is a potentially significant paper for a number of reasons. First, it provides another example of how the epidermis shapes dendrite patterning. Second, the function of the epidermis appears primarily mechanical through attachment of the vertex that is pulled behind the migrating cell rather than through some signaling mechanism. Third, a polarity protein (PAR-6) may be involved in localizing cell adhesion molecules to the leading tip of dendrites in the vertex. Fourth, several cell adhesion molecules function redundantly to potentially mediate this interaction between the epidermis and the dendrite tips. A further strength of the manuscript is the high resolution time lapse imaging of early embryonic processes of neuronal development in C. elegans, an aspect that remains, with few exceptions, significantly understudied. A weakness of the paper is that the focus of action for the cell adhesion molecules (HMR-1 and SAX-7) has not been established, and is only inferred from protein localization (which may not allow identification of the cellular source). The site of action for these factors could be important, because the cell adhesion molecules could constitute part of the receptor (complex) on either the dendrites or the epidermis, which interacts with the DYF-7 extracellular matrix protein. Regardless of possible physical interactions between these proteins, a few simple rescue experiments could allow the authors to substantially refine their model figure where, at this stage, the localization of HMR-1 and SAX-7 remains somewhat diffuse as it pertains to the phenomenon of dendrite extension by epidermal pull.

We appreciate reviewer #1’s nice summary and synthesis of our work. We agree with the reviewer that the site of action of the adhesion molecules is an important point for further understanding of the mechanisms. A major technical difficulty in the rescue experiments is the low penetrance of the dendrite anterior extension defect. In single mutant/RNAi, the penetrance is only a few percent, which makes the statistics of rescue problematic. In triple mutant/RNAi, the penetrance is in the 20-30% range, which is workable number-wise. However, a rescue in a triple mutant background may not be readily interpretable, given potentially complex cis- and trans-interactions among adhesion and receptor complexes. As a relatively minor point, we obtained a new CRISPR-based GFP tag of DLG-1. This new marker showed that DLG-1 is also localized to the dendrite tips (in addition to the edge of the hyp cells). We updated the text and Figure 3.

Other points:• Subsection “PAR-6 functions in amphid dendrite extension”. The conclusion that PAR-6 is required for attachment of dendrites to the epidermis rests on a single embryo. Moreover, the genotype of this one embryo is not clear and only inferred based on the assumption that all dead embryos are par-6 mutant embryos. I would consider this rather weak evidence, although maybe promising.

We agree with the reviewer. Reviewer #2 also raised a similar point. We were not able to collect a large enough number of embryos to substantiate the phenotype, but it may be useful for the community to document the observation. Therefore, we moved the text to the end of the section and the corresponding figure panel to supplement.

• The absence of an analysis of dex-1 is rather curious, given the known role of this gene with dyf-7 in retrograde dendrite extensions.

We thank the reviewer for pointing out this gap. We have added the analysis of *dex-1(ns42)*. This allele showed similar but weaker phenotype than *dyf-7(m537)*: it affected both the anterior extension and the retrograde extension, both at lower penetrance. We have updated the results in the text as well as in Figure 3.

• The authors make it sound as if the mechanism they describe goes against the previous model, but it really only refines the initial aspects of this process (third paragraph of subsection “Amphid neurons form a multicellular rosette before dendrite extension”).

We agree with the reviewer – we do not believe that our results go against the previous model, but rather show that retrograde extension is a multi-step process and employs different mechanisms at different stages. We have revised this passage ("These results show that the dendrites extend in two distinct steps: first, anterior-directed growth that extrudes the nascent dendrites from the neuronal cell bodies and brings the dendrite endings to the sensory depression at the nose, followed by posterior-directed growth that is concomitant with movement of the neuronal cell bodies."). We have also added a more extensive treatment in the Discussion ("Thus, retrograde extension of amphid dendrites occurs in at least two steps…") to clarify how our results relate to previous work on retrograde extension, as well as to possibly similar phenomena in *C. elegans* and vertebrates.

Reviewer #2:Bao and colleagues studied the initiation of sensory dendrites in vivo using the C. elegans amphid as a system. They described a surprising and exciting finding of a rosette structure where the dendrites of many neurons initiate. The rosette correlates and most likely is organized by a subcellular adhesion structure between the epidermal cell and the developing neurites. The migration of the epidermal cells drag the dendrite along which helps the first step of dendrite extension. The observation of the rosette is striking, original and extremely interesting. This observation provides an example of in vivo dendrite outgrowth that has not been described before. However, the paper is preliminary due to the incompletely characterization of this phenomenon. Several parts of the story can be improved.1) Figure 2 needs to be quantified. In the description of this figure, it was not immediately clear whether the anterior migration of the epidermal cell is normal or not. This needs to be carefully investigated and quantified.

We thank the reviewer for pointing out this essential dimension in interpreting our results, both in this section and below. We have added the analysis in the text. We divide the embryos into three groups based on the epidermal phenotype: 24 out of 79 embryos showed high expression of *lin-26*::GFP and more or less WT epidermal migration (weak effect of *elt-1(RNAi)*); in these embryos, dendrite extension reached the sensory depression as in WT. 29 out of 79 embryos showed low expression of *lin-26*::GFP and partial epidermal migration (stopping before covering the head completely; moderate effect of *elt-1(RNAi)*); in these embryos, the dendrites also made partial extension, stopping where the epiderm stopped and failing to reach the sensory depression. Finally, 26 out of 79 embryos showed minimal or no expression of *lin-26*::GFP (strong effect of *elt-1(RNAi)*); in these embryos, the amphid neurons remained in a rosette and showed minimal or no extension. We were not able to directly observe epidermal cell positions in these embryos due to the strong loss of *lin-26*::GFP, but by inference of RNAi strength and *lin-26*::GFP expression, we assumed that the epidermal migration defect must be more severe than the second group that showed partial migration. We considered other markers to assess the epiderm. Any epiderm specific maker (downstream of epidermal fate) would have the same technical issue as *lin-26*::GFP. A ubiquitous marker could bypass the labeling problem under *elt-1(RNAi)*, but it would be difficult to discern the leading edge of the epiderm.

2) Figure 3, in the triple mutants, is the anterior migration of the epidermal cells affected or is it just the dendrite extension that is defective?

We assessed the epiderm in two ways. First, did the embryo reach the 1.5-fold stage in time? Second, when one amphid showed dendrite extension defect, was the other amphid normal? [Given the low penetrance of our phenotype, the other side should be normal if epidermal migration in the given embryo was normal.] When reporting the dendrite phenotype, we only analyzed embryos that were deemed normal in terms of epidermal migration. For example, in *dyf-7; sax-7; dlg-1(RNAi)* embryos, 6 out of 72 arrested before reaching the 1.5-fold stage and were excluded from the statistics. We have updated the text and Figure 3 to reflect these details.

3) For Figure 4, is there a statistical test that the authors can do to test the significance of the various phenotypic categories? It seems that the par-6 results are very hard to interpret despite the very nice localization. The loss of function analyses is hampered by early defects and very low penetrance of the dendrite extension defect. I suggest removing this part of the data.

Please see above regarding the dendrite extension defect. We also added quantification of the DYF-7 signal to show that the area of DYF-7 localization is significantly larger than the WT.

4) For the second part of Figure 4, while the area of DYF-7 is detectably larger in the par-6 mutants compared with the control, the DYF-7 is nevertheless localized to the tip of the dendrite in the mutant. It is not clear what this phenotype really means.

We agree with the reviewer that the mechanism of DYF-7 localization, and more broadly the function of *par-6*, are still unclear. Heiman and Shaham (2009) showed that DYF-7 is first produced as a membrane protein. The extracellular domain is then cleaved to form part of the extracellular matrix. Our GFP tag labels the extracellular domain, so presumably the larger area means more diffuse DYF-7 ECM. However, it is not yet known whether the intact DYF-7 is localized to the rosette center before the extracellular domain is cleaved, or whether the extracellular domain is cleaved first and then somehow is recruited to the rosette center. The function of *par-6* in the whole process may be complex. For example, during the revision we found that DLG-1 is also localized to the dendrite tips, along with HMR-1. This observation raises the possibility that PAR-6 organizes a small apical domain at the rosette center/dendrite tips. One could imagine that DYF-7 may interact with this apical domain via several different mechanisms. The final answer would reveal interesting cell biology. However, as the reviewer pointed out, the technical constraints on *par-6* loss of function make the study difficult at the moment. New reagents and tools, such as better controlled PAR-6 degradation, would be needed. We have added this discussion to the manuscript.

[Editors' note: further revisions were requested prior to acceptance, as described below.]

The manuscript has been improved but there are some remaining issues that need to be addressed before acceptance, as outlined below:Major:– Please subject your data to more complete statistical analyses. For example, in Figure 3, the comparisons of the single, double and triple mutants to wild type animals are missing. The analyses should also include corrections for multiple comparisons.

The comparisons of all the experiment groups to wild type were performed and P values were listed in Figure 3—source data 1. For multiple comparisons, we did Bonferroni Corrections to determine the P value significance threshold.

– While the reviewers agreed that rescue may be difficult, they (and the editors) feel strongly that it is not sufficient to derive conclusions from single allele usage, particularly if no rescue is provided. Hence, you are asked to generate a dyf-7;sax-7 double mutant using different alleles from different labs and test this strain with and without RNAi against e.g. hmr-1 or dlg-1.

We obtained *dyf-7(ns119)* and *sax-7(eq1)* alleles and generated double mutants, as well as performed suggested experiments. We confirmed that *dyf-7(ns119); sax-7(eq1); hmr-1(RNAi)* and *dyf-7(ns119); sax-7(eq1); dlg-1(RNAi)* show significant higher defects in dendrite anterior extension (Figure 3—figure supplement 1B).